

# Unilateral application of an external pneumatic compression therapy improves skin blood flow and vascular reactivity bilaterally

Jeffrey S. Martin[1,2], Allison M. Martin[1], Petey W. Mumford[2], Lorena P. Salom[2], Angelique N. Moore[1] and David D. Pascoe[2]

[1] Department of Biomedical Sciences, Edward Via College of Osteopathic Medicine—Auburn Campus, Auburn, AL, United States of America
[2] School of Kinesiology, Auburn University, Auburn, AL, United States of America

## ABSTRACT

**Background.** We sought to determine the effects of unilateral lower-limb external pneumatic compression (EPC) on bilateral lower-limb vascular reactivity and skin blood flow.

**Methods.** Thirty-two participants completed this two-aim study. In AIM1 ($n = 18$, age: $25.5 \pm 4.7$ years; BMI: $25.6 \pm 3.5$ kg/m$^2$), bilateral femoral artery blood flow and reactivity (flow mediated dilation [FMD]) measurements were performed via ultrasonography at baseline (PRE) and immediately following 30-min of unilateral EPC treatment (POST). AIM2 ($n = 14$, age: $25.9 \pm 4.5$; BMI: $27.2 \pm 2.7$ kg/m$^2$) involved 30-min unilateral EPC ($n = 7$) or sham ($n = 7$) treatment with thermographic bilateral lower-limb mean skin temperature (MST) measurements at baseline, 15-min of treatment (T15) and 0, 30 and 60-min (R0, R30, R60) following treatment.

**Results.** Comparative data herein are presented as mean $\pm$ 95% confidence interval. AIM1: No significant effects on total reactive hyperemia blood flow were observed for the treated (i.e., compressed) or untreated (i.e., non-compressed) leg. A significant effect of time, but no time*leg interaction, was observed for relative FMD indicating higher reactivity bilaterally with unilateral EPC treatment (FMD: $+0.41 \pm 0.09$% across both legs; $p < 0.05$). AIM2: Unilateral EPC treatment was associated with significant increases in whole-leg MST from baseline during (T15: $+0.63 \pm 0.56$ °C in the visible untreated/contralateral leg, $p < 0.025$) and immediately following treatment (i.e., R0) in both treated ($+1.53 \pm 0.59$ °C) and untreated ($+0.60 \pm 0.45$ °C) legs ($p < 0.0125$). Across both legs, MST remained elevated with EPC at 30-min post-treatment ($+0.60 \pm 0.45$ °C; $p < 0.0167$) but not at 60-min post ($+0.27 \pm 0.46$ °C; $p = 0.165$). Sham treatment was associated with a significant increase in the treated leg immediately post-treatment ($+1.12 \pm 0.31$ °C; $p < 0.0167$), but not in the untreated leg ($-0.27 \pm 0.12$ °C). MST in neither the treated or untreated leg were increased relative to baseline at R30 or R60 ($p > 0.05$). Finally, during treatment and at all post-treatment time points (i.e., R0, R30 and R60), independent of treatment group (EPC vs. sham), there was a significant effect of region. The maximum increase in MST was observed at the R0 time point and was significantly ($p < 0.05$) larger in the thigh region ($+1.02 \pm 0.31$ °C) than the lower-leg ($+0.47 \pm 0.29$ °C) region. However, similar rates of MST decline from R0 in the thigh and lower leg regions were observed at the R30 and R60 time points.

Corresponding author
Jeffrey S. Martin,
jmartin@auburn.vcom.edu

**Discussion**. Unilateral EPC may be an effective intervention for increasing skin blood flow and/or peripheral conduit vascular reactivity in the contralateral limb. While EPC was effective in increasing whole-leg MST bilaterally, there appeared to be a more robust response in the thigh compared to the lower-leg. Thus, proximity along the leg may be an important consideration in prospective treatment strategies.

## INTRODUCTION

Chronic wound care therapies often target improvements in local blood flow. However, one of the difficulties in the treatment of chronic wounds is the pain and complication associated with direct wound contact. Interventions aiming to circumvent this problem that, in part, aim to improve blood flow and healing in chronic wounds include vacuum-assisted closure, local tissue warming and non-contact radiant bandages. However, the potential for a whole-leg, dynamic, external pneumatic compression (EPC) therapy applied to a non-affected limb has not been explored. We have previously demonstrated that a single treatment with a low pressure ($\leq$100 mmHg) EPC device (NormaTec Pro, NormaTec, Newton Center, MA, USA) acutely improves reactivity in the peripheral conduit and resistance vasculature of both compressed (i.e., legs) and non-compressed (i.e., arms) limbs (*Martin, Borges & Beck, 2015a*). Moreover, we have shown that local concentration of the stable metabolites of nitric oxide (NO) are increased in skeletal muscle biopsy samples following a 1-h bout of EPC (*Kephart et al., 2015*). Nevertheless, it has yet to be determined what effect unilateral EPC treatment would have on the contralateral limb. Recently, *Credeur et al. (2017)* showed that unilateral treatment for 1-h with intermittent pneumatic compression (IPC) at target inflation pressures of 120 mmHg in spinal cord injury patients acutely increases posterior-tibial artery reactivity in the treated (compressed) leg, but not the untreated leg. However, the aforementioned EPC device used in our prior investigations differs markedly from IPC as it is more dynamic (peristaltic compression pattern), utilizes a different duty cycle, contains additional inflation chambers and encompasses the entire leg. Given the need for non-contact and easily accessible interventions that can effectively improve blood flow and endothelial health to facilitate chronic wound healing, as a first step, we sought to determine the effect of dynamic, whole-leg EPC applied to a single leg on vascular reactivity and blood flow in both legs in a healthy population. We hypothesized that unilateral EPC treatment would effectively improve vascular reactivity, as determined by flow mediated dilation (FMD; AIM1), and skin microcirculation, as determined by dynamic infrared thermography mean skin temperature (MST; AIM2), in both the treated (i.e., compressed/covered) and untreated (i.e., non-compressed/non-covered) legs. In addition, we hypothesized that the improvements in skin blood flow in the limbs would be global (i.e., not segmental) in nature.

## MATERIALS & METHODS

### Participants

Twenty ($N = 20$) and fourteen ($N = 14$) apparently healthy persons were recruited from the local community to participate in AIM1 and AIM2 of this study, respectively. All procedures herein were approved by Auburn University's Institutional Review Board (Project #15-017 MR1502) and conformed to the standards set by the latest revision of the Declaration of Helsinki. Written informed consent was obtained from all participants prior to their voluntary participation in the study.

### Study design

#### *AIM1*

For AIM1, twenty persons (10 females, 10 males) reported for a single visit which was comprised of baseline bilateral blood pressure and popliteal artery FMD measurements, 30-min of unilateral EPC application with target inflation pressures of 70 mmHg, and a second set of bilateral FMD measurements 30-min after completion of the EPC treatment (POST). Bilateral blood pressure measures were employed to identify any significant between arm variability (defined as $\geq 10$ mmHg for systolic or diastolic blood pressure) that may confound the results (*Martin et al., 2015b*). In AIM1, two persons were identified as having significant between arm blood pressure variability and were excluded from the study. Thus, the total number of participants for AIM1 was eighteen (nine females, nine males).

#### *AIM2*

For AIM2, fourteen participants (four females, 10 males) reported for a single visit which was comprised of baseline bilateral blood pressure and leg thermography measurements, randomization to either 30-min of unilateral EPC treatment with target inflation pressures of 70 mmHg ($n = 7$) or a 30-min sham treatment with unilateral positioning of an EPC "leg sleeve" with no inflation/compression ($n = 7$), and 60-min of post-EPC/sham thermography measurements. Medical histories were assessed to confirm absence of medical conditions that might interfere with the intervention(s) or outcomes (e.g., orthopedic injuries/surgeries, neurological issues, diabetes, etc.). All participants were found to have regular thermal patterns in and between the two legs per preliminary imaging and were free from significant between arm blood pressure differences.

We have observed (informally) that the ultrasound/FMD procedure creates a slight, transient (<5 min) increase in skin surface temperature specific to the area of contact. Thus, separate trials were performed for each aim of the study.

### Procedures

For both aims, all participants were instructed to abstain from exercise and alcohol for 24-h and from caffeine for 12-h. Additionally, all participants reported to the laboratory at least 2-h post-prandial and, in order to control for any potential diurnal variation, at the same time of day.
### Brachial artery blood pressure

For both aims, upon reporting to the laboratory, height and weight was measured using a digital scale with height rod. Thereafter, following 15-min of supine rest, heart rate and brachial artery systolic and diastolic blood pressure measurements were made at both arms using an automated oscillometric device (OMRON BP785; Omron Corporation, Kyoto, Japan). The average of the measurements at the left and right arm were recorded for characterization of blood pressure.

### Flow mediated dilation

In AIM1, following baseline heart rate and blood pressure measurement, popliteal artery FMD was measured in both legs at baseline (i.e., PRE) and 30-min after (i.e., POST) unilateral EPC application. The order of FMD measurement side (i.e., left or right leg) at baseline was randomized for each subject, though that order was repeated for post-EPC measures.

FMD was used to determine endothelial-dependent reactivity in the popliteal arteries using high resolution ultrasound (GE Logiq S7 R2 Expert; GE Healthcare, Little Chalfont, UK) with a 3 to 12 MHz multi-frequency linear phase array transducer. In addition, the reactive hyperemia response (i.e., peak and total reactive hyperemia blood flow) during FMD procedures was determined for evaluation of microvascular (i.e., arterioles and capillaries) reactivity. In brief, each artery was imaged longitudinally with the transducer placed 3-8 cm below the popliteal fossa and held manually in the same position for the duration of the measurements. Simultaneous measurement of artery diameter and blood velocity was performed using duplex mode imaging (B-mode and Doppler) and video was captured through a digital interface at 30 frames/second with real time analysis (FMD Studio v2.8; QUIPU, Pisa, Italy). Resting measurements were captured for 1-min, reactive hyperemia was produced by inflating a cuff placed on the calf for 5-min at 200 mmHg, and, after cuff release, measurements were made for another 3-min. Vessel diameters were determined via automatic edge detection software (FMD Studio) measuring the distance between the near and far wall of the intima. Blood velocity was determined via selection of a region of interest around the Doppler waveform and a trace of the velocity-time integral was used to calculate mean velocity for each cardiac cycle. Shear rate was calculated as [(4*time averaged mean velocity)/vessel diameter] and mean blood flow was calculated as [$\Pi$* (diameter/2)$^2$* time average mean velocity * 60].

FMD was calculated as absolute (aFMD; in mm) and relative (%FMD) peak change in artery diameter in response to the hyperemic stimulus. Because dilation also depends on the resultant reactive hyperemia shear rate stimulus, normalized FMD (nFMD) was calculated as the ratio of %FMD to the shear rate area under the curve (AUC) to the time at which the maximal diameter was observed (100*%FMD/shear rate AUC). Finally, peak reactive hyperemia blood flow and total reactive hyperemia blood flow AUC were calculated from observed 5-s averages during the reactive hyperemia period. AUC was determined using the sum of trapezoids method after baseline values were subtracted.

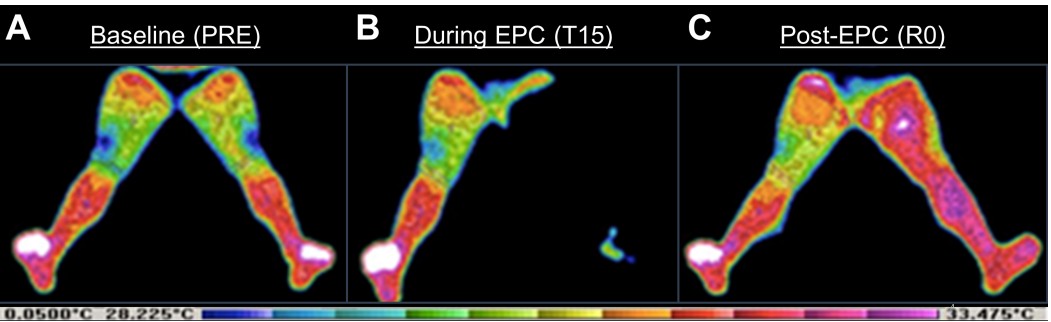

**Figure 1** **Representative thermograms.** Pictured are thermograms of a participant's lower limbs (A) after equilibration to environmental conditions (i.e., baseline [PRE]), (B) during external pneumatic compression (EPC; T15), and (C) immediately post-EPC (R0).

### Thermography assessment of skin blood flow

In AIM2, following 15-min of rest for equilibration to room/position conditions, infrared images of the anterior aspect of the legs were taken using an infrared thermal imaging system camera and processor (Computerized Thermal Imaging Inc., Ogden, UT, USA). Regions of interest included the anterior thigh and lower-leg. Leg region MST was determined from the processor program that provides the average value of the pixels within the regions of interest. Images were taken within the thermal lab where the temperature was maintained at $21 \pm 1.1\,°C$ (thermoneutral environment) and emissivity was set at 0.98. Images were taken at baseline, at the mid-point of EPC/sham treatment (T15), immediately post-EPC/sham treatment (R0), and at 30- (R30) and 60-min (R60) following EPC/sham treatment. All participants wore gym shorts and non-constrictive undergarments appropriate for complete visualization of the thighs. A representative thermography image is presented in Fig. 1.

### External pneumatic compression (EPC) and sham

For both AIM1 and AIM2, a dynamic, sequential EPC device (NormaTec Pro, Newton Center, MA, USA) was applied for 30-min. Briefly, the EPC device utilizes "leg sleeves" which contain five circumferential inflatable chambers (arranged linearly) encompassing the leg from the feet to the hip/groin. The "leg sleeves" are connected to an automated pneumatic pump at which duration of treatment and target inflation pressures can be controlled. In this study, we chose to use target inflation pressures of ~70 mmHg for each chamber as we've previously observed acute improvements in vascular reactivity with bilateral treatment at this target inflation pressure (*Martin, Borges & Beck, 2015a*). Herein, only one leg sleeve was applied (right leg) while the contralateral leg (left) was uncovered. At the onset of the EPC treatment, all zones in the "leg sleeve" are pre-inflated to form fit the participant's legs while applying minimal pressure (~10 mmHg). Thereafter, the distalmost chamber (which covers from the high ankle to toes) inflated to approximately 70 mmHg while the remaining zones were not inflated. For 30-s, this chamber "pulsed" after which the pressure was held constant at 70 mmHg and the same process occurred in the zone above (calves) for another 30-s. Following this 30-s, the ankle zone deflated

completely, the calf remained at a constant pressure of approximately 70 mmHg, and the process occurred in the next highest zone (high claves/lower thighs). This continued until the highest zone (5th zone/upper thighs) finished the process at which point all zones were completely deflated for 30-s and a full compression cycle (3-min total) was completed. This entire cycle of compression was repeated continuously over the course of the treatment session which lasted for 30-min.

In AIM2, a sham condition was also employed which consisted of 30-min application of a single EPC "leg sleeve" on the right leg, but devoid of any actual compression. A sham was included in AIM2 to assess the thermogenic effect from contact and/or attenuation of heat loss from simply wearing the "leg sleeve".

### Statistical analysis

All statistical analyses were performed using SPSS v22.0 (IBM Corp., Armonk, NY, USA). Prior to statistical analysis normality was confirmed using Shapiro–Wilk tests. For ANOVA, if sphericity was violated a correction factor was applied to hypothesis testing. Subject characteristics for both aims were compared using independent $t$-tests.

For AIM1 dependent variables, two-way repeated measures ANOVAs were performed with time (PRE vs. POST) and leg (treated vs. untreated) as independent variables. When a significant time*leg interaction was observed, *post-hoc* comparisons were made using Bonferroni corrected paired $t$-tests.

For AIM2, MST during treatment (i.e., T15) was analyzed using a repeated measures of analysis with time (PRE vs. T15), region (thigh vs. lower-leg), and treatment group (EPC vs. sham) as the independent variables. Given the opaque nature of the leg sleeve, leg (i.e., treated vs. untreated) was not included as an independent variable as MST during respective treatments was only evaluable in the untreated leg. MST at R0, R30 and R60 was analyzed using repeated measures ANOVA with time (PRE vs. R0, R30 and R60), leg, region, and treatment group as the independent variables. When significant interactions with time were found, *post-hoc* comparisons were made using paired and independent t-tests as appropriate with Bonferroni correction for multiple comparisons (0.05/number of comparisons). Data are presented as mean $\pm$ standard deviation for summary data and as mean (95% confidence interval (lower limit, upper limit)) for comparative data.

## RESULTS

### Subject characteristics

Participant characteristics for the cohorts of subjects included in AIM1 and AIM2 are presented in Table 1. In AIM1, significant between sex differences existed for height, body mass, and blood pressures, but not age ($p = 0.211$) and BMI ($p = 0.123$). In AIM2, no significant between condition differences (i.e., EPC vs. sham) were found for any characteristic ($p > 0.05$). Significant between sex differences existed for height ($p = 0.014$), body mass ($p = 0.040$), and diastolic blood pressure ($p = 0.039$), but not systolic blood pressure ($p = 0.130$), age ($p = 0.457$) or BMI ($p = 0.203$). However, sex was not found to be a significant covariate in either AIM1 or AIM2.

**Table 1  Participant characteristics for AIM1 and AIM2.**

|  | AIM1 | | | AIM2 | | |
|---|---|---|---|---|---|---|
|  | Overall ($N = 18$) | Males ($n = 9$) | Females ($n = 9$) | Overall ($N = 14$) | EPC ($n = 7$) | Sham ($n = 7$) |
| Age, yrs | 25.5 ± 4.7 | 26.9 ± 4.8 | 24.1 ± 4.3 | 25.9 ± 4.5 | 25.6 ± 4.0 | 26.2 ± 5.3 |
| Height, m | 1.70 ± 0.09 | 1.76 ± 0.8 | 1.64 ± 0.05[**] | 1.74 ± 0.09 | 1.75 ± 0.07 | 1.73 ± 0.10 |
| Body mass, kg | 75.1 ± 16.0 | 84.4 ± 16.1 | 65.7 ± 9.6[**] | 82.9 ± 14.9 | 85.8 ± 12.7 | 79.9 ± 17.3 |
| BMI, kg/m² | 25.6 ± 3.5 | 26.9 ± 3.5 | 24.3 ± 3.2 | 27.2 ± 2.7 | 28.0 ± 2.4 | 26.4 ± 2.9 |
| LA SBP, mmHg | 112 ± 10 | 117 ± 9 | 108 ± 9[*] | 117 ± 10 | 117 ± 10 | 117 ± 12 |
| LA DBP, mmHg | 71 ± 6 | 76 ± 4 | 66 ± 3[**] | 74 ± 6 | 77 ± 3 | 72 ± 8 |
| RA SBP, mmHg | 111 ± 11 | 116 ± 10 | 105 ± 10[*] | 115 ± 13 | 117 ± 11 | 114 ± 15 |
| RA DBP, mmHg | 71 ± 8 | 75 ± 7 | 66 ± 8[*] | 74 ± 8 | 77 ± 5 | 71 ± 10 |

Notes.
Data are mean ± SD.
BMI, body mass index; LA, left arm; SBP, systolic blood pressure; DBP, diastolic blood pressure; RA, right arm.
**, * $p < 0.05$ and $p < 0.01$, respectively, for between sex comparisons (AIM1).

**Table 2  Popliteal artery flow-mediated dilation characteristics at baseline and after EPC in the treated and non-treated leg.**

|  | EPC ($n = 18$) | |
|---|---|---|
|  | PRE | POST |
| **Treated Popliteal Artery** | | |
| Resting diameter, mm | 5.41 ± 0.96 | 5.49 ± 0.94 |
| Resting MBV, cm/s | 7.7 ± 1.9 | 7.4 ± 1.9 |
| Resting mean shear rate, s⁻¹ | 58.2 ± 14.0 | 55.3 ± 15.2 |
| Reactive hyperemia MBV AUC, cm/s | 300 ± 119 | 243 ± 121[*] |
| Reactive hyperemia shear rate AUC, A.U. | 2,245 ± 897 | 1,878 ± 1,204[*] |
| Absolute FMD, mm | 0.11 ± 0.04 | 0.13 ± 0.05[*] |
| **Untreated Popliteal Artery** | | |
| Resting diameter, mm | 5.12 ± 0.92[†] | 5.16 ± 0.86[†] |
| Resting MBV, cm/s | 8.0 ± 1.2 | 7.1 ± 1.6[*] |
| Resting mean shear rate, s⁻¹ | 64.6 ± 14.9 | 56.2 ± 13.8[*] |
| Reactive hyperemia MBV AUC, cm/s | 331 ± 129 | 266 ± 121[*] |
| Reactive hyperemia shear rate AUC, A.U. | 2,688 ± 1,147 | 2,121 ± 1,026[*] |
| Absolute FMD, mm | 0.09 ± 0.03 | 0.12 ± 0.04[*] |

Notes.
Data are expressed as mean ± SD.
Mean blood velocity (MBV) and shear rate were measured continuously and resting measures are representative of the average of 2 min of baseline measurements, whereas area under the curve (AUC) measures are representative of values following cuff release until maximal dilation was observed; FMD is flow mediated dilation expressed as absolute (mm), relative (%), and normalized to the reactive hyperemic shear rate AUC. * and **, significantly difference between time points (pre vs. post; ($P < 0.05$ and $P < 0.01$, respectively); †, significantly different between legs at the same time point (e.g., post in compressed artery vs. post in non-compressed artery; $P < 0.05$).

## AIM1—effect of unilateral EPC on bilateral vascular reactivity in the legs

Results from popliteal artery FMD assessment are presented in Table 2 and Fig. 2. No main effects (time or treatment) or interaction was observed for resting popliteal artery diameter or blood flow (Table 2; $p > 0.05$ for all). With respect to the reactive hyperemia

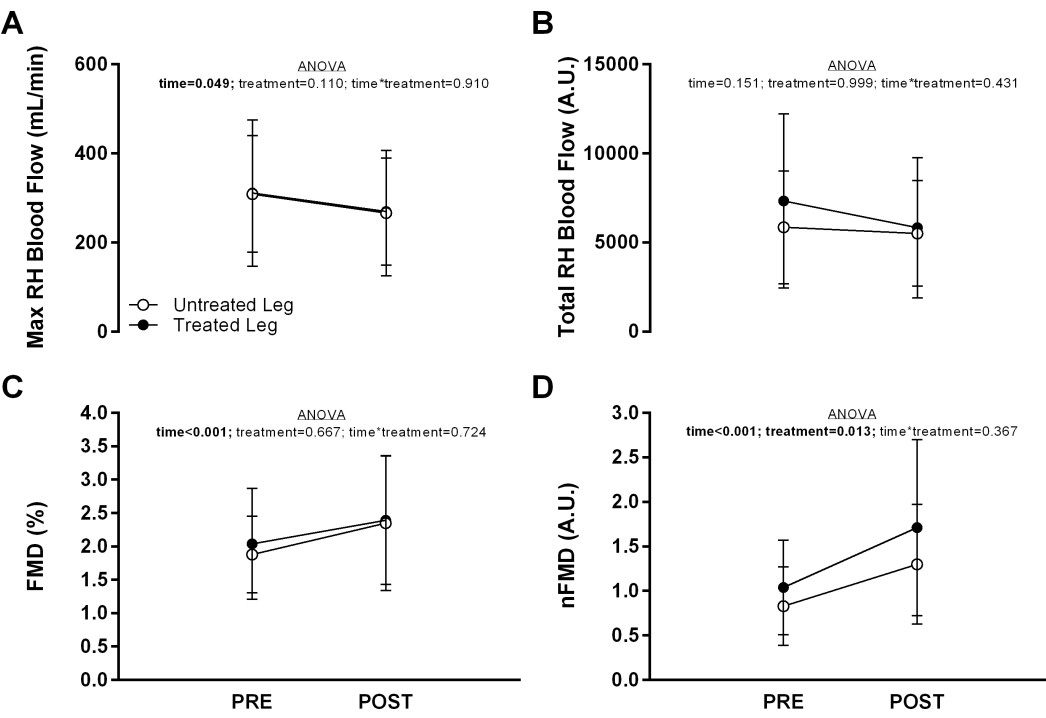

**Figure 2** **Popliteal artery blood flow and flow mediated dilation (FMD) characteristics in both legs before (PRE) and 30-min after (POST) a bout of unilateral external pneumatic compression (EPC).** (A) maximal (max) popliteal artery blood flow during reactive hyperemia, (B) total popliteal artery blood flow during reactive hyperemia, (C) relative FMD (%), and (D) FMD normalized to shear rate area under the curve to peak dilation (nFMD). Data are presented as mean ± SD for each leg (treated vs. untreated) at each time point (PRE vs. POST). ANOVA results are presented within each panel with significant main effects emphasized in bold.

response, a significant main effect of time, but no main effect of leg or time*leg interaction, was observed for both maximum blood flow (Fig. 2A) and shear rate AUC to peak dilation ($p < 0.010$; Table 2). Independent of leg (collapsing across treated and untreated legs), maximum reactive hyperemia blood flow and shear rate AUC at the POST time point were decreased relative to baseline (i.e., PRE) by 39.3 mL/min [−0.9, −77.8] and 467 a.u. [−734, −200], respectively. No main effects or interactions were observed for total reactive hyperemia blood flow (Fig. 2B).

A significant effect of time ($p < 0.001$), but no main effect of leg or time*leg interaction, was observed for aFMD and pFMD. Independent of leg, both aFMD (+0.022 mm [+0.009, +0.035]; Table 2) and pFMD (+0.41% [+0.14, +0.68]; Fig. 2C) were significantly higher at the POST time point relative to PRE. For nFMD, significant main effects of time and leg were observed, but no interaction (Fig. 2D). Independent of time (collapsing across PRE and POST time points), nFMD was significantly higher in the treated leg (+0.31 [+0.10, +0.53]) and independent of leg, nFMD was significantly higher at POST (+0.57 [+0.39, +0.80]).

**Table 3** Average and regional mean skin temperature (MST) at baseline in each leg.

| | Right leg | | | Left leg | | |
|---|---|---|---|---|---|---|
| | Thigh MST | Lower-leg MST | Average MST | Thigh MST | Lower-leg MST | Average MST |
| EPC ($n = 7$) | $30.6 \pm 1.1$ | $31.1 \pm 0.8$ | $30.9 \pm 1.0$ | $31.0 \pm 0.9$ | $31.0 \pm 1.1$ | $31.0 \pm 1.0$ |
| Sham ($n = 7$) | $30.6 \pm 0.5$ | $31.2 \pm 0.7$ | $30.9 \pm 0.6$ | $30.5 \pm 0.5$ | $31.2 \pm 0.7$ | $30.9 \pm 0.6$ |

**Notes.**
Data are mean $\pm$ SD. EPC, external pneumatic compression.

### AIM2—effect of unilateral EPC on bilateral leg skin temperature

Baseline (PRE) values for regional and average MST in each leg for each treatment group are presented in Table 3. For during treatment (T15), significant time*treatment ($p = 0.003$) and time*region ($p = 0.031$) interactions were observed. Across both regions, change in MST at T15 was significantly greater in the untreated leg with EPC compared to sham ($+0.82\,°C$ [$+0.34, +1.30$]; Fig. 3A). MST in the untreated leg was significantly elevated relative to baseline with EPC treatment EPC and significantly reduced relative to baseline with sham in the untreated leg. Across both treatments, change in MST was significantly greater in the thigh region compared to the lower-leg region ($+0.32\,°C$ [$+0.13, +0.51$]; Fig. 3B). However, collapsing across both treatments, neither the thigh ($p = 0.051$) or lower-leg ($p = 0.731$) region were significantly different from baseline.

For R0, significant region*time ($p < 0.001$) and treatment*leg*time ($p = 0.032$) interactions were observed. Fig. 3D illustrates the effect of region where, when collapsing across both treatments and legs, MST was significantly elevated from baseline at R0 at both regions ($p < 0.001$ and $p = 0.003$ for thigh and lower-leg, respectively) though the increase was significantly greater in the thigh compared to the lower-leg ($+0.55\,°C$ [$+0.40, +0.70$]; $p < 0.001$). With respect to the treatment*leg*time interaction (illustrated in Fig. 3C), independent of region, EPC treatment was associated with a significant increase in MST in both the treated leg ($p < 0.001$) and untreated leg ($p = 0.002$) at R0 whereas MST was increased with sham in the treated leg ($p < 0.001$) but decreased in the untreated leg ($p = 0.002$). Change in MST at R0 in the treated leg with EPC treatment was significantly higher than in the untreated leg with EPC treatment ($+0.94\,°C$ [$+0.56, +1.31$]; $p < 0.001$) and the untreated leg with sham treatment ($+1.81\,°C$ [$+1.24, +2.37$]; $p < 0.001$), but not the treated leg with sham treatment ($+0.42\,°C$ [$-1.05, +0.21$]; $p = 0.076$). Change in MST at R0 in the treated leg with sham was significantly higher than in both the untreated leg with EPC ($+0.52$ [$-0.03, +1.07$]; $p = 0.012$) and the untreated leg with sham ($+1.39\,°C$ [$+1.12, +1.65$]; $p < 0.001$). Finally, in the untreated leg, change in MST at R0 was greater with EPC treatment compared to sham treatment ($+0.87\,°C$ [$+0.35, +1.38$]; $p < 0.001$)

For the R30 time point, significant treatment*time ($p < 0.001$), leg*time ($p = 0.001$) and region*time ($p < 0.001$) interactions were observed. Across both regions and legs, MST was significantly elevated relative to baseline with EPC treatment ($p < 0.001$) and significantly reduced relative to baseline with sham treatment ($p = 0.003$; Fig. 4A). Across both treatments and legs, thigh MST was significantly elevated relative to baseline ($p < 0.001$), but not lower-leg MST ($p = 0.558$; Fig. 4B). Finally, across both regions

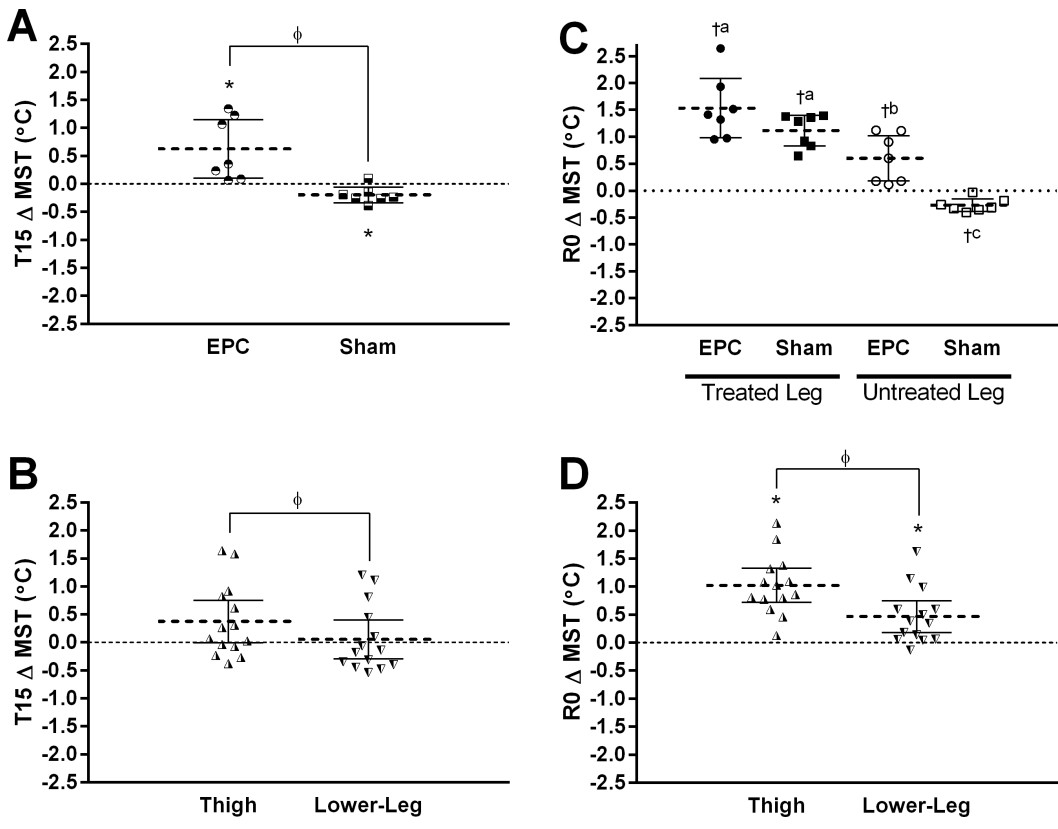

**Figure 3 Change (Δ) from baseline in mean skin temperature (MST) at the mid-point (T15) and immediately following (R0) external pneumatic compression (EPC)/sham treatment.** Data are presented as individual values and mean ± 95% confidence interval. Panels illustrate the observed (A) time*treatment (EPC vs. Sham; data collapsed across regions) and (B) time*region (thigh vs. lower-leg; data collapsed across treatments) interactions for MST at T15, and (C) time*treatment*leg (treated vs. untreated; data collapsed across regions) and (D) time*region (data collapsed across legs and treatments) interactions for MST at R0. *, significant change from baseline ($p < 0.0167$) and $\phi$, significantly different between factors ($p < 0.0167$). For panel c: †, significant change from baseline ($p < 0.0125$) and different superscript letters between groups indicate significant differences ($p < 0.008$).

and treatments, treated ($p = 0.034$) and untreated ($p = 0.873$) leg MST were no longer significantly elevated relative to baseline (Fig. 4C).

Similar to the R30 time point, at the R60 time point, significant treatment*time ($p = 0.002$), leg*time ($p = 0.008$) and region*time ($p < 0.001$) interactions were observed. Across both regions and legs, MST was no longer significantly elevated relative to baseline with EPC treatment ($p = 0.051$), but remained significantly reduced relative to baseline with sham treatment ($p < 0.001$; Fig. 4D. Across both treatments and legs, thigh MST was also no longer significantly elevated relative to baseline ($p = 0.101$), but now significantly reduced relative to baseline in the lower-leg MST ($p < 0.001$; Fig. 4E). Finally, across both regions and treatments, treated leg ($p = 0.492$) and untreated leg ($p = 0.075$) MST remained unchanged relative to baseline (Fig. 4F).

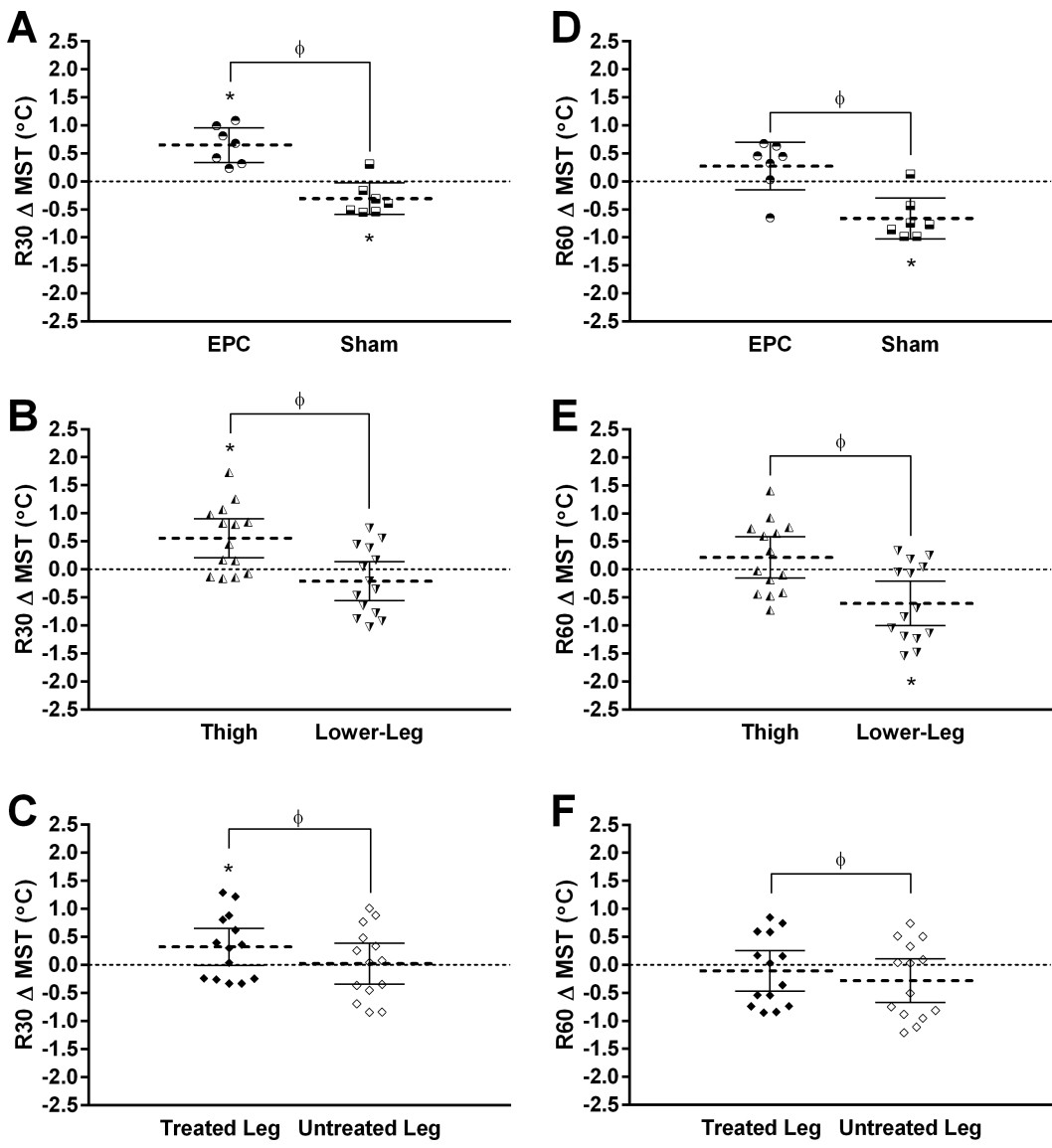

**Figure 4 Change from baseline in mean skin temperature (MST) at 30- (R30) and 60-min (R60) following external pneumatic compression (EPC)/sham treatment.** Data are presented as individual values and mean ± 95% confidence interval. Panels illustrate the observed (A) time*treatment (EPC vs. Sham; data collapsed across regions and legs), (B) time*region (thigh vs. lower-leg; data collapsed across treatments and legs), and (C) time*leg (treated vs. untreated; data collapsed across treatments and regions) interactions for MST at R30 and (D) time*treatment, (E) time*region, and (F) time*leg interactions for MST at R30. *, significant change from baseline ($p < 0.0167$) and $\phi$, significantly different between factors ($p < 0.0167$).

## DISCUSSION

The primary findings of the present study are as follows; (1) 30-min of unilateral EPC is associated with increased popliteal artery FMD in both the treated and untreated leg post-EPC, (2) MST during treatment is significantly increased relative to baseline in the

untreated leg with EPC treatment, but not sham treatment, (3) MST is increased in both treated and untreated legs immediately following EPC treatment, but only in the treated leg with sham treatment, (4) the increase in MST in the treated leg was more transient with sham treatment compared to EPC treatment, and (5) independent of other factors, the thigh region was associated with more robust and sustained increases in MST compared to the lower-leg.

Regarding AIM1 of the present investigation, the bilateral improvement in popliteal artery FMD with unilateral treatment is similar to our previous observations of improved popliteal (compressed leg) and brachial artery (non-compressed arm) FMD with 1-h of bilateral leg EPC treatment (*Martin, Borges & Beck, 2015a*). Herein, we extend our previous findings by demonstrating that (1) unilateral EPC improves vascular reactivity in the treated and untreated (contralateral) leg, (2) 30-min of treatment, compared to 1-h, is sufficient for improvement of peripheral conduit reactivity, and (3) FMD is improved for at least 30-min following treatment. Improved FMD within the treated leg could be explained by markedly altered hemodynamic shear stress, and resultant local NO bioavailability, during EPC treatment (*Martin et al., 2016*), but this is an unlikely mechanism in the contralateral leg. However, a single 1-h bout of IPC has been shown to affect systemic measures of NO as well as non-local vascular reactivity (*Rifkind et al., 2014*). While IPC does differ markedly from the EPC therapy used herein, it is not unreasonable to assume that other external compression stimuli can modulate changes in non-local and/or systemic NO bioavailability. We have also previously posited that EPC-mediated myokine (e.g., IL-6) release may potentiate systemic NO bioavailability (*Fisslthaler & Fleming, 2009*; *Horman et al., 2008*; *Pedersen & Febbraio, 2008*), but this remains unknown. Moreover, while NO bioavailability may play a role in the observed acute FMD increase in the untreated leg, a decrease in sympathetic nervous system (SNS) efferent activity may also be involved in the observed response with EPC treatment (*Dyson, Shoemaker & Hughson, 2006*; *Hijmering et al., 2002*; *Lind, Johansson & Hall, 2002*). Indeed, the pressor effect from static compression on the lower limb(s), at a range of pressures (30–90 mmHg), and with single limb compression, has been shown previously and is abolished by epidural anesthesia (*Williamson et al., 1994*). Moreover, we have previously observed acute increases in peripheral (brachial) and central (aortic) blood pressure for at least 10-min following bilateral EPC treatment (*Vincent-Horta et al., 2015*). Thus, it is possible that the arterial baroreflex may modulate sympathetic outflow in response to acute EPC treatment. However, the role of the SNS in the acute FMD response to EPC requires further, specific, investigation.

In regards to AIM2 of the present investigation, thermography measures temperature changes within the cutaneous microcirculation plexus that are modulated by skin blood flow perfusion. As such, infrared imaging provides a non-contact thermal map of the skin surface area that reveals both spatial and temporal changes that occur in the regional temperature distribution (*Pascoe et al., 2012*). Notably, although infrared imaging provides accurate thermal measures of skin surface temperatures that are responsive to blood flow alterations, it cannot quantify blood flow itself.

In the treated legs, immediately following EPC/sham treatment (i.e., R0) a similar, significant rise in temperature of greater than 1 °C was observed with both EPC (+1.53 °C)

and sham ($+1.12\,°C$). The EPC "leg sleeve" is a non-breathable, thick material for which conductive contact did not allow for heat transfers (*Havenith, 1999*), likely mediating a significant proportion of the increase in MST of the treated leg with both EPC and sham. Indeed, while the change in MST was greater in the treated leg with EPC compared to sham ($+0.42\,°C$) it was not a statistically significant difference ($p = 0.073$). However, in the untreated leg, immediately following completion of sham/EPC treatment (i.e., R0), MST was significantly elevated relative to baseline with EPC ($+0.60\,°C$), but significantly decreased relative to baseline with sham ($-0.27\,°C$) illustrating a markedly different response to the two treatments. This was similar to what was observed during treatment (i.e., T15), where there was a significant increase across both regions in the untreated leg with EPC ($+0.83\,°C$), but a significant decrease with sham ($-0.20\,°C$). Thus, it would appear that during and immediately following unilateral treatment, the addition of a pneumatic compression stimulus not only prevents a decrease in MST in the contralateral limb, but actually promotes a significant increase. Although skin sympathetic nerve activity (*Wallin, Sundlöf & Delius, 1975*) and sweat rate (*Wilson, Cui & Crandall, 2001*) do not change with the aforementioned arterial baroreflex, changes in cutaneous vascular conductance have been observed (*Crandall et al., 1996*; *Kellogg, Johnson & Kosiba, 1990*; *Tripathi & Nadel, 1986*). Regardless, one explanation could simply be the distribution, or re-distribution, of heat and/or blood in the vascular space (*Arndt et al., 1985*). Indeed, the higher MST in the contralateral leg with EPC, but not sham, during and immediately following treatment could be due to heat added to the system from the dynamic compression stimulus (i.e., tissue deformation) and/or compression itself mobilizing fluid(s) including, but not limited to, a proportion of venous and arterial flow.

Differential responses between treatments (EPC vs. sham) were also observed during the recovery period from EPC/sham treatment (i.e., R30, R60). Indeed, 30-min following completion of unilateral treatment, independent of leg, MST remained elevated respective to baseline with EPC and was actually reduced relative to baseline with sham treatment. In addition, 60-min following treatment, independent of leg, while MST had returned to baseline with EPC treatment, it remained reduced relative to baseline with sham. The absence of a significant interaction with the factor of leg (treated vs. untreated) suggests an additional effect of unilateral dynamic compression on the bilateral thermal recovery response. In the treated leg, transmural pressure changes in the vasculature of the compressed tissue are likely associated with an auto-regulatory (i.e., myogenic) response which may augment skin blood flow and MST. Importantly, this effect has been shown to persist for hours following pressure application (*Bochmann et al., 2005*). However, devoid of a compression stimulus, this is unlikely to be a major contributor to the contralateral limb. Thermal control over cutaneous vascular responses has not been fully elucidated but have affirmed roles for temperature sensitive afferent neurons, sympathetic/parasympathetic controls and NO mediated vasodilation (*Kellog Jr, 2006*) which may have, in some capacity, contributed to our observations. While speculative, acute massage, to which lower limb EPC is often compared, has been shown to elicit a decrease in SNS efferent activity and an increase in parasympathetic nervous system activity (*Diego & Field, 2009*). Moreover, twenty 40-min sessions of massage therapy over

the course of 4-weeks has been shown to decrease autonomic nerve conduction latency and amplitude (*Lee, Park & Kim, 2011*). However, future studies are warranted with respect to autonomic nervous system modulation and its role in local and non-local responses to EPC treatment.

Several findings herein would appear to be of interest from a clinical standpoint. First, acute improvements in bilateral popliteal artery FMD would suggest improved peripheral vascular reactivity which may potentiate greater blood flow and wound healing (*Guo & Dipietro, 2010*). In addition, improved endothelium dependent and independent vascular reactivity may decrease the risk for diabetic foot ulcer development (*Dinh et al., 2012*). Secondly, although we did not observe any change following the EPC condition in peak or total reactive hyperemia assessed with ultrasonography during the FMD procedure, this is similar to our previous findings with EPC (*Martin, Borges & Beck, 2015a*). Notably, in that study, despite no changes in reactive hyperemia assessed via ultrasonography, we did identify an acute increase in peak reactive hyperemia blood flow responses immediately following an EPC condition when assessed via venous occlusion plethysmography. Finally, local blood flow improvements may mitigate negative ischemic, inflammatory, oxidative, and bacterial associated factors in wound healing (*Guo & Dipietro, 2010*; *Hopf et al., 1997*; *Kurz, Sessler & Lenhardt, 1996*; *Mustoe, 2004*). Herein, we observed marked increases in MST, a surrogate of skin blood flow, in the contralateral leg during and for at least 30-min following unilateral EPC treatment. Thus, non-contact treatment via compression of the unaffected limb may be a viable strategy to improve skin blood flow during wound healing. Importantly, at all time points (during and after treatment) significant effects of region were observed with a more robust increase in thigh MST being observed initially with a subsequently similar rate of decline over the recovery period. Thus, the regional nature of the responses needs to be considered. In addition, from an experimental model perspective, our findings suggest treatment differences may be disguised when using a one leg (control) model in which the untreated leg is altered by the intervention. In our case, the increased temperatures in the "control" leg lessened the calculated statistical effect between legs. Finally, we used target inflation pressures of ~70 mmHg only for the EPC intervention and the response to higher or lower target inflation pressure(s) may vary.

## CONCLUSIONS

Unilateral, dynamic, whole-leg EPC treatment increases bilateral vascular reactivity and MST. With respect to MST, it is important to recognize that although there is a greater and more persistent effect in the treated leg with MST, single leg treatment will also provide a clinically significant increase in the contralateral limb. Moreover, EPC treatments of the leg were strongly influenced by the intervention response in the thigh, and to a lesser extent the lower-leg. From a clinical perspective, our results suggest that EPC could be utilized to increase blood flow in an afflicted limb (e.g., injury, peripheral vascular disease, etc.) with contralateral application, particularly when the target area is more proximal in the leg.

## ACKNOWLEDGEMENTS

The authors wish to thank the participants for their compliance and for devoting their time to this study.

### Funding

The authors received no funding for this work.

### Competing Interests

The authors declare there are no competing interests.

### Author Contributions

- Jeffrey S. Martin and David D. Pascoe conceived and designed the experiments, performed the experiments, analyzed the data, contributed reagents/materials/analysis tools, prepared figures and/or tables, authored or reviewed drafts of the paper, approved the final draft.
- Allison M. Martin, Lorena P. Salom and Angelique N. Moore performed the experiments, authored or reviewed drafts of the paper, approved the final draft.
- Petey W. Mumford performed the experiments, analyzed the data, prepared figures and/or tables, authored or reviewed drafts of the paper, approved the final draft.

### Human Ethics

The following information was supplied relating to ethical approvals (i.e., approving body and any reference numbers):

The Auburn University Institutional Review Board granted approval to carry out the study within its facilities (Project #15-017 MR1502).

### Data Availability

The raw data are provided in Supplemental Information 1.

### Supplemental Information

Supplemental information for this article can be found online at http://dx.doi.org/10.7717/peerj.4878#supplemental-information.

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
