# Peer review of "Unilateral application of an external pneumatic compression therapy improves skin blood flow and vascular reactivity bilaterally"

_PeerJ, doi:10.7717/peerj.4878_

## Round 0.1 · original submission · Minor Revisions

The reviewers have raised several questions and comments to clarify and strengthen the argument that is made in the manuscript. Please follow their suggestions to improve the revised manuscript.

Reviewer 1 ·

Basic reporting

The paper is well written, and easy to follow.

Experimental design

The experimental design is sound, and data appear to have been carefully collected. There are a few lines needing clarification:

In the study design, line 92-95, was the 30-min post FMD measures performed immediately following the EPC bout, or 30 mins later (i.e., 60 mins from start of EPC)?

Lines 102-105, was the sham simply a control, or were EPC cuffs positioned and not inflated? I know it is indicated elsewhere in the Methods, but please clarify here too.

Lines 142-143, include the version and company name for the FMD Studio software, I’m assuming QUIPU.

Validity of the findings

The authors back-up previous work from their laboratory in regards to the peripheral vascular changes following EPC. However, there are some recent reports that would be worth referencing in the manuscript:

Lines 174-176, what was the rationale for utilizing an EPC pressure of 70 mmHg? A previous report (Credeur et al. J Spinal Cord Med, 2017) used a pressure of 120 mmHg (Intermittent Pneumatic Compression—IPC), and found a localized improvement in leg vascular health. Is there a threshold pressure for improvements? Please elaborate, either in this section or in Discussion.

There is a fairly recent report (Rifkind et al. Nitric Oxide, 2014) which investigated the impact of leg IPC on systemic changes in NO-mediated markers, such as plasma nitrite, and vascular function in the arms. It would be worth incorporating this reference into your discussion of bilateral changes in leg FMD in response to unilateral EPC.

Additional comments

No further comments.

Reviewer 2 ·

Basic reporting

This is an interesting study on physiological response to local therapeutic tissue compression. The background may be, according to the authors, clinical requirements. Had it been like that, more justification would be needed. In the present form it is not convincing, at least for the reviewer who is a clinician. It might be better to focus attention to purely physiological phenomena during local tissue compression.

Experimental design

The presented measuring methods have been chosen correctly and described sufficiently. What is lacking is a vague description of application of the compression sleeve. Were chambers inflated sequentially, how fast, distal to proximal pressure gradient? What was the skin temperature immediately after removal of the sleeve as it goes down within 1-3 minutes after exposure to ambient temperature. Another words, how much heat was produced by the inflating sleeve at the chamber-skin interface (mechanical to heat). How much heat energy was supplied to the investigated person by compression? How much heat energy was retained in the part of limb covered by the sleeve for 30 minutes? All this is easy to calculate. Heat generation and distribution process?

Validity of the findings

Compression resulting in tissue deformation and mobile tissue fluid mobilization provides energy which is fast dissipated all over the body due to blood perfusion and brings about a systemic response to heat energy input. Subsequently, heat excess is being eliminated in other parts of the body exposed to temperature lower than that of core tissues. No wonder that increase in temperature through dilated skin microvasculature in the contralateral was observed. This type of "consensual" reaction has been known for long.

Additional comments

The problem of increase of tissue elasticity during therapeutic limb tissue compression has been widely discussed. No consensus has been reached. Some consider it to be caused by increase of tissue temperature (mechanical to heat energy) , others calculated forces and found that total heat retained in tissues is negligible as it is immediately washed out.
The reviewer suggests:
1. Introduction: focus widely on physiology and not clinical aspects
2. Methods: add more detailed description of compression
3. Results: try to calculate the heat energy balance during compression
4. Discussion: concentrate upon heat balance basing on the relevant contemporary literature

---

## Round 0.2 · accepted · Accept

The authors have adequately addressed all reviewer comments.